# Topical Absorption of Glutathione–Cyclodextrin Nanoparticle Complex in Healthy Human Subjects Improves Immune Response against *Mycobacterium avium* Infection

**DOI:** 10.3390/antiox12071375

**Published:** 2023-07-02

**Authors:** Kayvan Sasaninia, Melissa Kelley, Arbi Abnousian, Ali Badaoui, Logan Alexander, Nisar Sheren, James Owens, Shlok Rajurkar, Brianna Razo-Botello, Abraham Chorbajian, Sonyeol Yoon, Sanya Dhama, Edith Avitia, Cesar Ochoa, Ray Yutani, Vishwanath Venketaraman

**Affiliations:** 1College of Osteopathic Medicine of the Pacific, Western University of Health Sciences, Pomona, CA 91766, USA; kayvan.sasaninia@westernu.edu (K.S.); arbi.abnousian@westernu.edu (A.A.); ali.badaoui@westernu.edu (A.B.); logan.alexander@westernu.edu (L.A.); nisar.sheren@westernu.edu (N.S.); jimowens333@gmail.com (J.O.); abraham.chorbajian@westernu.edu (A.C.); sonyeol.yoon@westernu.edu (S.Y.); ryutani@westernu.edu (R.Y.); 2Graduate College of Biomedical Sciences, Western University of Health Sciences, Pomona, CA 91766, USA; melissa.kelley@westernu.edu; 3Division of Biological Sciences, University of California Berkeley, Berkeley, CA 94720, USA; shlok.rajurkar@berkeley.edu; 4College of Natural and Agricultural Science, University of California Riverside, Riverside, CA 92521, USA; brazo002@ucr.edu; 5Keck Science Department, Pitzer College, Claremont, CA 91711, USA; sdhama@students.pitzer.edu; 6WesternU Center for Clinical Research, Western University of Health Sciences, Pomona, CA 91766, USA; eavitia@westernu.edu (E.A.); cochoa@westernu.edu (C.O.)

**Keywords:** glutathione, oxidative stress, mycobacterium, cyclodextrin, topical

## Abstract

Glutathione (GSH) is an important intracellular antioxidant responsible for neutralizing reactive oxygen species (ROS). Our laboratory previously demonstrated that the oral administration of liposomal GSH improves immune function against mycobacterium infections in healthy patients along with patients with HIV and Type 2 diabetes. We aim to determine if the topical application of a glutathione–cyclodextrin nanoparticle complex (GSH-CD) confers a therapeutic effect against mycobacterium infections. In our study, healthy participants received either topical GSH-CD (n = 15) or placebo (n = 15) treatment. Subjects were sprayed four times twice a day for three days topically on the abdomen. Blood draws were collected prior to application, and at 1, 4, and 72 h post-initial topical application. GSH, malondialdehyde (MDA), and cytokine levels were assessed in the processed blood samples of study participants. Additionally, whole blood cultures from study participants were challenged with *Mycobacterium avium (M. avium)* infection in vitro to assess mycobacterium survival post-treatment. Topical GSH-CD treatment was observed to elevate GSH levels in peripheral blood mononuclear cells (PBMCs) and red blood cells and decrease MDA levels in PBMCs 72 h post-treatment. An increase in plasma IL-2, IFN-γ, IL-12p70, and TNF-α was observed at 72 h post-topical GSH-CD treatment. Enhanced mycobacterium clearance was observed at 4 h and 72 h post-topical GSH-CD treatment. Overall, topical GSH-CD treatment was associated with improved immune function against *M. avium* infection. The findings of this pilot study suggest GSH–cyclodextrin complex formulation can be used topically as a safe alternative mode of GSH delivery in the peripheral blood.

## 1. Introduction

Glutathione (GSH) is an intracellular thiol produced by cells to detoxify free radical species during metabolism, infection, or environmental exposure [1]. GSH is a tripeptide consisting of glutamate, glycine, and cysteine and maintains redox homeostasis by serving as a reducing agent against reactive oxygen species (ROS) [2,3]. Increasing evidence suggests GSH serves important functions in maintaining immune health, including its role in the immune response against *Mycobacterium tuberculosis* (*M. tb*) infection [4,5]. *M. tb* is the causative agent for tuberculosis (TB) and accounts for approximately 6.5 million infections worldwide yearly [6]. Immunocompromised patients such as those with HIV and Type 2 diabetes mellitus (T2DM) are at increased risk for contracting *M. tb* [7,8,9].

GSH deficiency found in patients with HIV and T2DM is caused by the diminished production of GSH-synthesizing enzymes [7,8,9]. Furthermore, GSH deficiency is associated with increased oxidative stress and susceptibility to mycobacterial infections [4,5]. Methods of GSH delivery have posed a challenge as the oral or intravenous administration of free GSH yields low bioavailability [10,11]. Our lab has previously demonstrated that healthy patients along with patients with HIV and Type 2 diabetes experience an increase in GSH levels, a reduction in oxidative stress, and improved immune function when GSH is carried out within liposomes and administered orally [12,13]. Topical formulations of GSH have been studied extensively on humans and animal models for their dermatological effects and exist commercially as a skin-brightening agent [14,15,16]. However, the effects of topical GSH application on modulating GSH levels and immune function have yet to be explored.

In this study, we aim to test whether the topical application of GSH cyclodextrin complex (GSH-CD) can be absorbed in the skin and increase GSH levels in the peripheral blood. Furthermore, we aim to assess if GSH-CD confers a therapeutic effect against mycobacterium infections using the *Mycobacterium avium (M. avium)* whole-blood infection model. The findings of this study can offer insight into alternative GSH delivery methods and adjunctive therapeutic approaches in controlling mycobacterium infections.

## 2. Materials and Methods

### 2.1. Study Design

This study is a randomized, double-blind, placebo-controlled clinical trial conducted at the Western University of Health Sciences and approved by the Western University of Health Sciences Institutional Review Board, protocol # FB22/IRB/005. All participants in the trial were provided with written informed consent before study enrollment. Additionally, the study is in the registry of ClinicalTrials.gov (Identifier: NCT05926245). Eligible participants were randomly assigned to either the GSH-CD arm (n = 15) or the placebo arm (n = 15) of the study. Blood draws occurred prior to and after topical treatment. A portion of the collected blood samples were processed to separate plasma, red blood cells (RBCs), and peripheral blood mononuclear cells (PBMCs) for further assessment of relevant biomarkers. Another portion of the blood samples were used for whole blood infection assays. Auro Pharmaceuticals Inc. manufactured GSH-CD consisting of an rGSH cyclodextrin complex (200 mg/mL rGSH). Additionally, the same company manufactured and supplied the placebo solution consisting of empty cyclodextrin nanoparticles. All components in the GSH-CD (GSH, ascorbic acid, cyclodextrin, potassium sorbate, and radish root extracts) or the placebo (ascorbic acid, cyclodextrin, potassium sorbate, and radish root extracts) are considered generally regarded as safe by the FDA. An overview of the study is demonstrated in Figure 1.

### 2.2. Subject Selection, Clinical Encounters, Treatment Application

Healthy patients between the ages of 21 and 65 were recruited at the Patient Care Center (PCC) at the Western University of Health Sciences. Upon confirmation and consent, the subjects met with a physician to conduct a physical examination and blood draw to ensure the participants were healthy and met the eligibility criteria. The blood samples were prescreened with a comprehensive metabolic panel, including an assessment for the presence of HIV antibodies, surface hepatitis B antigen, HbA1C levels, and (if applicable) a pregnancy test. The exclusion criteria included health conditions and lifestyle choices that can modify levels of GSH, ROS, and inflammatory markers in the body [7,8,9,12,13]. The exclusion criteria involved patients with active cancer, uncontrolled T2DM, HIV/AIDS, hepatitis, TB, or COVID-19. Additionally, patients with a history of alcohol abuse within the past 6 months, pregnancy, or allergies to any of the components of the solution were excluded from the study. Patients were recruited and screened until the placebo and GSH-CD arms reached the target sample size.

Screened healthy patients meeting inclusion criteria had blood drawn to establish pretreatment baseline 30 min prior to topical solution application. The participants topically applied 4 sprays of either GSH-CD or placebo on the ventral abdomen and had their blood drawn 1 h and 4 h after the initial application. The participants continued to apply 4 sprays (0.5 mL or 100 mg rGSH) of the topical solution twice a day (6 h apart) until the final blood draw 72 h after the initial application. All patients were financially compensated by the end of the study and gained access to their complete metabolic panel.

### 2.3. Blood Processing and Storage

Blood samples were collected and processed to isolate plasma, red blood cells (RBCs), and peripheral blood mononuclear cells (PBMCs) from whole blood. Blood component isolation was achieved via density centrifugation using ficoll histopaque (Sigma, St. Louis, MO, USA). Whole blood was gently placed over ficoll histopaque in a 1:1 ratio and centrifuged at 1800 rpm for 30 min at 25 °C. The plasma, RBCs, and PBMCs were aliquoted into microcentrifuge tubes and stored at −80 °C until further use.

### 2.4. GSH Quantification

Total and oxidized GSH levels were measured using the GSH colorimetric kit from Arbor Assays (Catalog No. K006-H1) following the manufacturer’s protocol (Arbor Assays, Ann Arbor, MI, USA). Reduced GSH levels were calculated by subtracting oxidized GSH concentration from total GSH concentration.

### 2.5. Malondialdehyde (MDA) Quantification

MDA levels were assessed using the Thiobarbituric Acid Reactive Substances (TBARS) assay kit from Cayman Chemicals (Item No. 10009055) following the manufacturer’s protocol (Cayman Chemical Company, Ann Arbor, MI, USA).

### 2.6. Cytokine Assessment

Levels of cytokines IL-12, IL-2, IFN-γ, and TNF-α in the plasma were assessed via sandwich ELISA using the Meso Scale Discovery V-Plex Plus Proinflammatory Panel kit (Catalog No: K15049G-1) following manufacturer’s protocols (MSD, Rockville, MD, USA).

### 2.7. Total Protein Quantification

Total protein in the RBCs and PBMCs of healthy patients was assessed via a Pierce BCA Protein Assay Kit (Catalog No. 23227) following the manufacturer’s protocols (Thermo Scientific, Rockford, IL, USA).

### 2.8. Bacterial Preparation

The *M. avium* ATCC 25291 strain was cultured in 7H9 medium (Hi Media, Santa Maria, CA, USA) supplemented with albumin dextrose complex (ADC) (GeminiBio, West Sacramento, CA, USA) and incubated at 37 °C and grown until reaching the logarithmic growth phase (indicated by an optical density between 0.5 and 0.8 at A600). *M. avium* cultures were centrifuged at 4000 rpm to form a pellet. The supernatant was discarded, and the bacterial pellet was washed with PBS (Sigma, St Louis, MO, USA) and subsequently disaggregated by vortexing five times with 3 mm sterile glass beads at 3 min intervals. *M. avium* suspension was then filtered using a 5 μm syringe filter to remove any remaining bacterial aggregations. The single-cell suspension of processed *M. avium* was serially diluted and plated on 7H11 agar (Sigma, St Louis, MO, USA) to determine the bacterial numbers present in the processed stock. Aliquots of processed bacterial stocks were frozen and stored in a cryogenic freezer at −80 °C. The frozen processed stocks of *M. avium* were thawed and used at the time of the experimental study.

### 2.9. Whole Blood Infection Assay

Whole blood samples from each time point of the blood draw were diluted with RPMI with 5% AB serum and inoculated with *M. avium* for a whole blood infection assay. Briefly, RPMI-diluted blood samples were infected in vitro with processed *M. avium* with a multiplicity of infection ratio of 1:10 (bacteria: PBMCs) in triplicate in a 96-well culture plate. *M. avium*-infected blood samples were incubated at 37 °C with 5% CO_2_ and terminated at 1 h and 72 h post-infection. The 1 h termination point signifies a sudden response to infection, and the 72 h point reflects sustained infection. During termination, cells from infected blood samples were lysed with cold sterile nanopure water to liberate phagocytosed mycobacterium. Cell lysates were serially diluted, plated on 7H11 agar, and incubated at 37 °C for 2 weeks. Colony-forming units (CFUs) were counted on each plate to assess bacterial burden and survival. An overview of the blood draws and infection termination time points. A summary of the whole blood infection assay is demonstrated in Figure 2.

### 2.10. Statistical Analysis

Statistical data analysis was performed using GraphPad Prism Software version 9. The levels of GSH, MDA, cytokines, and CFUs were compared between the placebo and treatment groups using an unpaired *t*-test with Welch’s correction. Multiple group comparison was performed using ordinary one-way ANOVA. Following data collection, data quality assessment and exclusion were conducted prior to group comparisons. Statistically significant differences between treatment groups were determined when the *p*-value < 0.05. A single asterisk (*) denotes a *p*-value < 0.05, and a double asterisk (**) indicates a *p*-value < 0.005.

## 3. Results

### 3.1. Elevated Levles of Reduced GSH after GSH-CD Topical Treatment

Reduced GSH (rGSH) levels were assessed in the RBCs of healthy patients between the GSH-CD group (n = 12) and the placebo group (n = 12) post-topical solution application. Sampling errors resulting in out-of-range rGSH levels from three participants in each arm were excluded from the analysis. rGSH is significantly elevated in the RBCs of healthy patients receiving GSH-CD 72 h post-initial application compared to the placebo group (Figure 3A). The rGSH levels were assessed in the PBMCs of healthy patients 72 h post-treatment between the placebo group (n = 12) and the GSH-CD group (n = 13). Out-of-range rGSH levels of three patients in the placebo group and two patients from the GSH-CD group were excluded from the analysis. rGSH was observed to be significantly elevated in the PBMCs of healthy patients 72 h post-GSH-CD application compared to the placebo (Figure 3B).

### 3.2. MDA Levels Are Decreased after Topical GSH-CD Treatment

Malondialdehyde (MDA) is a stable end product of lipid peroxidation and was measured to assess oxidative stress after topical treatment. Sampling errors resulting in out-of-range values were excluded. Levels of MDA in the PBMCs were measured 4 h post-initial treatment application between the GSH-CD arm (n = 12) and the placebo arm (n = 12). Out-of-range MDA levels in the PBMCs from three participants in each arm were excluded from the analysis. MDA was observed to be decreased in the PBMCs in healthy patients receiving GSH-CD treatment 4 h post-initial application compared to the placebo (Figure 4A). The MDA levels in the PBMCs were compared between participants in the placebo (n = 12) and GSH-CD (n = 12) groups. Out-of-range MDA levels from three participants in each arm were excluded from the analysis. A reduction in MDA was maintained at 72 h in patients receiving GSH-CD treatment compared to patients receiving the placebo (Figure 4B).

### 3.3. Increased Levels of IL-12, IL-2, IFN-γ and TNF-α in Plasma Post-Initial GSH-CD Treatment

We assessed the cytokine profiles of participants in between the placebo and GSH-CD groups after topical treatment. Sampling errors resulting in out-of-range values were excluded. The plasma levels of IL-2 were measured between the placebo group (n = 11) and the GSH-CD group (n = 11), with four patients with out-of-range values from each arm excluded from the analysis. IL-2 was elevated 72 h post-initial GSH-CD treatment (Figure 5A). Plasma IFN-γ levels were compared in the plasma (n = 14) and GSH-CD groups (n = 14) with one patient with out-of-range values from each arm excluded from the analysis. IFN-γ was observed to be elevated 72 h post-initial GSH-CD treatment in the plasma compared to the placebo treatment (Figure 5B). Plasma TNF-α levels were compared between the placebo group (n = 13) and the GSH-CD group (n = 15) after 72 h post-initial treatment. TNF-α levels from two participants with out-of-range values in the placebo group were excluded from the analysis. An increase in TNF-α levels was observed 72 h post-initial treatment in participants receiving GSH-CD treatment compared to the placebo (Figure 5C). Plasma IL-12p70 levels were compared between the placebo (n = 13) and GSH-CD (n = 13) groups. IL-12p70 levels from two participants from each arm were excluded from the analysis. Plasma IL-12p70 levels were elevated in the GSH-CD group 72 h post-initial treatment (Figure 5D).

### 3.4. Topical GSH-CD Treatment Is Associated with the Reduction in the Burden of M. avium (In Vitro)

*M. avium* survival in whole blood cultures derived from healthy patients receiving topical treatment 1 h and 72 h post-infection was compared between the placebo (n = 15) and GSH-CD (n = 15) groups 30 min prior, and 1, 4, and 72 h post-initial topical treatment. Blood cultures from samples before or 1 h post-topical treatment showed no significant difference in *M. avium* survival between the placebo or GSH-CD groups, 1 h post-*M. avium* infection (Figure 6A,B). Blood cultures from samples collected 4 h and 72 h post-treatment showed a significant reduction in *M. avium* survival in the GSH-CD group 1 h post-*M. avium* infection (Figure 6C,D). Blood cultures from samples collected before and 1 h post-topical treatment showed no significant differences in *M. avium* survival 72 h post-*M. avium* infection (Figure 6E,F). Blood cultures from samples collected 4 h and 72 h post-GSH-CD treatment showed a significant reduction in *M. avium* survival compared to the placebo 72 h post-*M. avium* infection (Figure 6G,H).

## 4. Discussion

GSH is a tripeptide that has two forms, the reduced form (rGSH) and the oxidized form (GSSG). GSH is synthesized through a two-step pathway. First, glutamate and cysteine are joined by a glutamine–cysteine ligase (GCL) [17]. Next, γ-glutamylcysteine and glycine are joined by a glutathione synthase (GSS) [18]. GSSG is the product of a GSH redox reaction with an ROS, with two equivalents of GSH conjoined with a disulfide bond. GSSG can be converted to two molecules of rGSH by the enzyme glutathione reductase (GSR), using NADPH as a cofactor [19]. rGSH possesses antioxidant capabilities and protects cells by serving as a cofactor for glutathione peroxidase to reduce free radicals and peroxides [20]. Due to its immunomodulatory properties, GSH can play a significant role in adjunct therapy for *M. tb* infection [4,21,22].

Methods of GSH delivery have been a topic of increasing interest due to the low bioavailability of free GSH and impaired GSH synthesizing capabilities in immunocompromised patients [9,10]. The efficacy of intravenous GSH has been inconclusive, possessing intrinsic risks associated with injection at the site of application [23]. The oral administration of GSH carried in liposomes has previously been shown to restore GSH levels, reduce oxidative stress, and improve host immune responses against *M. tb* in patients with HIV and T2DM [12,13,24]. However, patients with gastrointestinal problems could experience difficulty with oral supplementation, including mild side effects such as increased flatulence, loose stools, flushing, and weight gain [25].

In this study, we aim to assess if the topical application of the GSH–cyclodextrin nanoparticle complex, GSH-CD, can confer a noninvasive route of GSH delivery and a therapeutic effect against mycobacterium infections. Cyclodextrins are cyclic oligosaccharides with amphiphilic properties and have been increasingly used as a carrier for topical drugs due to their ability to enhance drug solubility and bioavailability [26]. GSH-CD consists of rGSH, ascorbic acid, cyclodextrin, potassium sorbate, and radish root extracts. The placebo treatment contained the same solution excluding the rGSH. Ascorbic acid was used to prevent the oxidation of rGSH in air, and radish root extracts were used to prevent contamination and for their antimicrobial properties. Cyclodextrin has previously been reported to possess antioxidant properties [27]. To assess whether the ascorbic acid and cyclodextrin in the formulation may elicit confounding antioxidant capabilities, we compared the MDA levels of the peripheral blood mononuclear cells (PBMCs) of the patients in the placebo group prior and 72 h post-initial topical application (Figure A1). MDA is the stable product of lipid peroxidation events due to cellular injury by ROS and thus serves as a measure of oxidative stress [28]. We did not observe any decrease in PBMC MDA levels from sham placebo treatment, indicating placebo treatment does not diminish cellular oxidative stress.

No adverse side effects were reported during or after the application. Following the GSH-CD treatment, the rGSH levels were observed to be increased in the red blood cells (RBCs) and the peripheral blood mononuclear cells (PBMCs) from samples collected 72 h post-initial GSH-CD application, indicating continued application for three days is necessary to achieve elevated GSH levels in the blood (Figure 3). Conversely, significantly lower MDA levels were observed in the PBMCs of GSH-CD-supplemented patients at 4 and 72 h post-initial treatment suggesting reduced oxidative stress in immune cells due to the increased availability of rGSH from the topical treatment (Figure 4).

GSH is known to stimulate the production of T-helper 1 (Th1) cytokines necessary for T cell activation and granulomatous responses against mycobacterial infections [7,12]. Th1 cytokines, such as IL-2, IFN-γ, and TNF-α, have been previously reported to be stimulated by GSH in uninfected macrophages and are responsible for the activation and cell recruitment of macrophages to the site of infection [7,21,29]. IL-12p70 is a cytokine produced by antigen-presenting cells, such as dendritic cells, which stimulate CD4 T cells to produce IFN-γ [30]. IFN-γ produced by Th1 lymphocytes activates macrophage function and differentiation [31]. Stimulated Th1 lymphocytes also produce IL-2 to maintain the viability of T cells and coordinate the generation of effector and memory cells [32,33]. TNF-α is a pleiotropic cytokine produced by macrophages and induces the apoptosis of infected tissues [34]. Additionally, TNF-α promotes the phagosome-lysosomal fusion of phagocytosed mycobacterium [35]. In the placebo group, we observe no significant difference in pretreated or placebo-treated plasma IFN-γ. A significant reduction in IL-2 production was observed along with a nonsignificant reduction in IL-12p70 and TNF-α by the study endpoint in the placebo group compared to pretreatment (Figure A2). Consistent with our previous study with L-GSH oral supplementation, GSH-CD treatment significantly elevated the levels of IL-12p70, TNF-α, IL-2, and IFN-γ 72 h after the initial treatment was administered compared to a placebo, indicating enhanced Th1 cytokine production after GSH-CD treatment (Figure 5).

A whole blood infection assay was performed to assess if GSH-CD topical treatment affects the control of *M. avium* infection (Figure 6). *M. avium* is one of the most common nontuberculosis mycobacteria to cause pulmonary infections [36]. *M. avium* is ubiquitous in the environment, and immunocompromised individuals are at an elevated risk of acquiring *M. avium* infection [37]. *M. avium* is treated with macrolides, such as azithromycin and clarithromycin, alongside other agents such as rifampicin [38]. However, the emergence of antibiotic-resistant strains presents the need to develop new treatment modalities. We previously demonstrated that L-GSH supplementation augments host protective effects against *M. tb* and *Mycobacterium bovis* strain Bacillus Calmette–Guérin (BCG); however, we have yet to assess if GSH is efficacious against *M. avium* [13,39]. With the observed elevation in rGSH, reduced oxidative stress, and the stimulation of Th1 cytokines after GSH-CD treatment, we proposed that GSH-CD could lower the burden of *M. avium* infection. GSH-CD-supplemented patients demonstrated significantly higher control of the *M. avium*-infected whole blood at 4 h and 72 h after the initial treatment when compared to the placebo group at both 1 h and 72 h post-infection. These findings suggest that GSH-CD treatment can yield improved control of the initial and sustained *M. avium* infection 4 h after the initial topical application. Continued GSH-CD application for three days yielded significantly reduced *M. avium* survival.

Overall, our findings suggest that topical GSH-CD application may improve the control of *M. avium* infection by increasing blood rGSH levels, reducing oxidative stress, and inducting a Th1-promoting cytokine response. Sampling errors resulting in the exclusion of participant data from analysis and reducing comparison group size present a limitation to this pilot study. Thus, further studies with increased sample sizes are needed to assess the efficacy of topical GSH-CD in elevating GSH levels in healthy or immunocompromised patients, such as those with HIV or T2DM. Topical GSH-CD treatment did not result in the complete clearance of *M. avium* infection by the study endpoint, warranting further studies evaluating optimal dose ranges, treatment durations, and combination treatment with first-line antibiotics. No adverse effects were reported during this study; however, an evaluation for the potential for skin discoloration or heightened TNF-a levels after the long-term application of GSH-CD is recommended. While *M. avium* infection in this study served as a model for mycobacterium infections in the blood, additional in vitro studies are needed to assess the efficacy of GSH-CD against *M. tb* infection. Additionally, placebo-controlled randomized trials with patients with active *M. tb* infections are needed to confirm the efficacy of topical GSH-CD as an adjunctive therapeutic approach against TB.

## 5. Conclusions

The topical application of the glutathione–cyclodextrin nanoparticle complex is a non-invasive route of administration for GSH delivery with no reported side effects in healthy patients. GSH-CD treatment may enhance the immune response, resulting in a reduction in the burden of *M. avium* in vitro. Improved *M. avium* control may be a result of increased rGSH levels, reduced oxidized stress, and increases in the Th1 cytokines IFN-γ, IL-12p70, IL-2, and TNF-α after GSH-CD treatment (Figure 7).

## Figures and Tables

**Figure 1 antioxidants-12-01375-f001:**
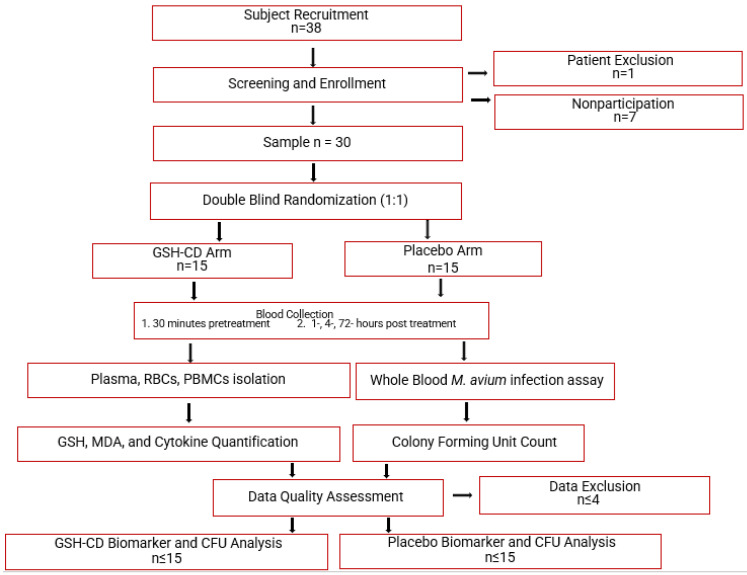
Topical glutathione cyclodextrin complex clinical trial study design. Subjects (n = 38) were screened for inclusion and exclusion criteria. After screening, 1 participant was excluded due to chronic diabetes and 7 eligible participants did not continue to complete the study before the initial blood draw. Participants (n = 30) were randomly assigned to either the placebo arm (n = 15) or the GSH-CD arm (n = 15) blind to both the participant and investigator. Participants were instructed to apply 4 sprays (0.5 mL) of topical GSH-CD solution (cyclodextrin with 100 mg rGSH) or placebo (cyclodextrin only) on the abdomen twice a day for 3 days. Blood draws were collected before and after topical solution application at various time points. Blood samples were processed to extract plasma, RBCS, and PBMCs to assess for relevant biomarkers. A portion of the blood sample was used to perform whole blood infection assays. Placebo and treatment arms were compared after data quality assessment and data exclusion.

**Figure 2 antioxidants-12-01375-f002:**
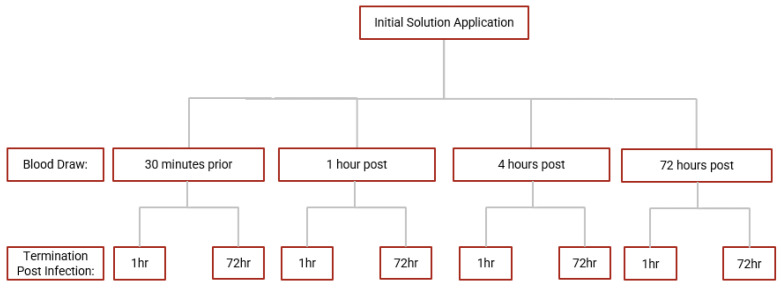
Scheme for whole blood infection assay. Whole blood derived from blood samples collected at each time point in the study were infected with *M. avium* and incubated until termination at 1 h and 72 h post-infection.

**Figure 3 antioxidants-12-01375-f003:**
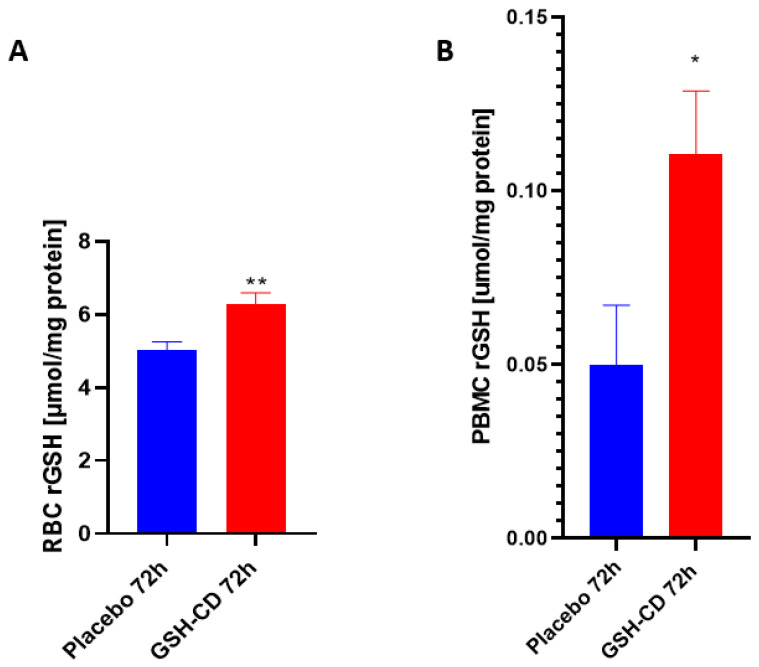
**Glutathione (GSH) levels in healthy patients receiving placebo or GSH-CD topical treatment.** (**A**) Levels of the reduced form of GSH in the RBCs isolated from the placebo (n = 12) and GSH-CD (n = 12) groups at 72 h post-initial application. (**B**) Levels of reduced GSH in the PBMCs isolated from the placebo (n = 12) and GSH-CD (n = 13) groups at 72 h post-initial application. Measurements for the rGSH levels are normalized against the total protein levels in the RBCs and PBMCs. All comparisons were made via an unpaired *t*-test with Welch’s correction. Single and double asterisks indicate statistical significance with *p*-values < 0.05 and <0.005, respectively.

**Figure 4 antioxidants-12-01375-f004:**
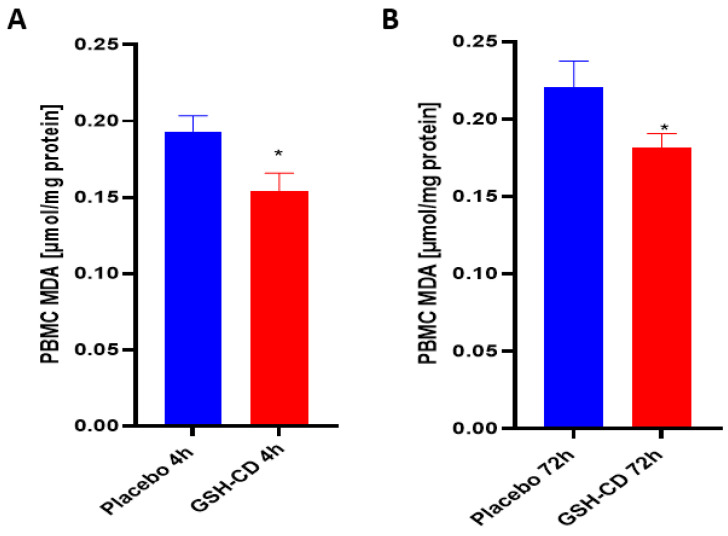
**Malondialdehyde (MDA) levels in healthy patients receiving placebo or GSH-CD topical treatment.** (**A**) MDA levels in the PBMCs isolated from the placebo (n = 12) and GSH-CD (n = 12) groups at 4 h post-initial application. (**B**) MDA levels in the PBMCs isolated from the placebo (n = 12) and GSH-CD (n = 12) groups at 72 h post-initial application. Measurements for the PBMC MDA are normalized against total PBMC protein levels. All comparisons were made via an unpaired *t*-test with Welch’s correction. Asterisks indicate statistical significance with a *p*-value < 0.05.

**Figure 5 antioxidants-12-01375-f005:**
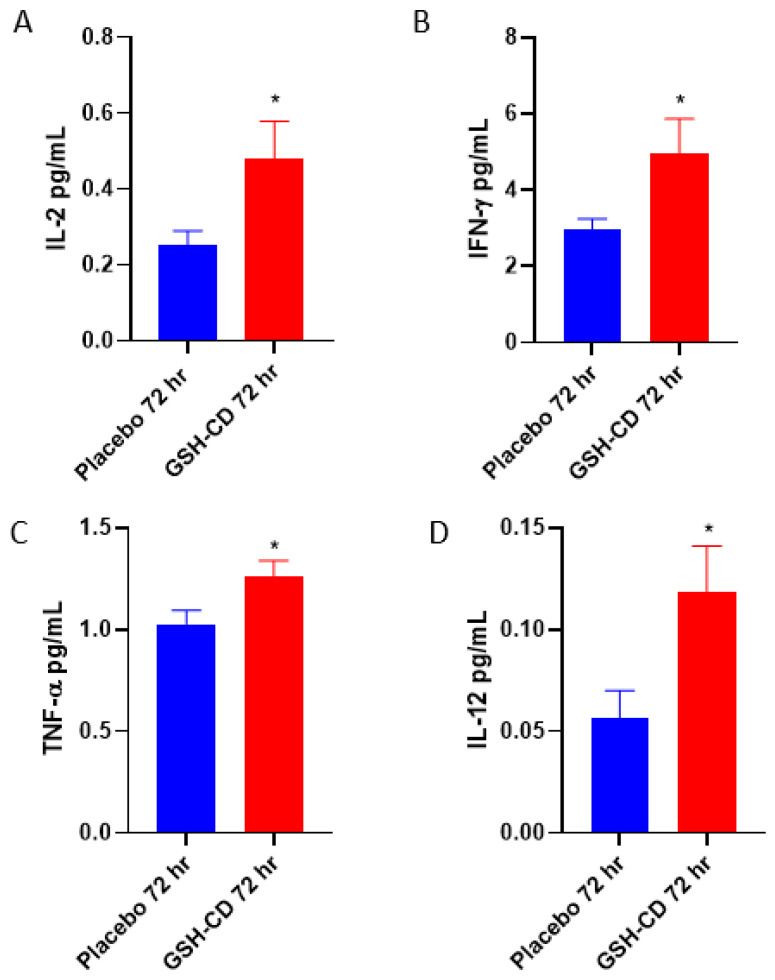
**Cytokine profile of healthy patients receiving placebo or GSH-CD treatment.** (**A**) Levels of IL-2 in the plasma isolated from the placebo (n = 11) and GSH-CD (n = 11) groups at 72 h post-initial application. (**B**) Levels of IFN-γ in the plasma isolated from the placebo (n = 14) and GSH-CD (n = 14) groups at 72 h post-initial application. (**C**) Levels of TNF-α in the plasma isolated from the placebo (n = 13) and GSH-CD (n = 15) groups at 72 h post-initial application. (**D**) Levels of IL-12 in the plasma isolated from the placebo (n = 13) and GSH-CD (n = 13) groups at 72 h post-initial application. All comparisons were made via unpaired *t*-test with Welch’s correction. Asterisks indicate statistical significance with a *p*-value < 0.05.

**Figure 6 antioxidants-12-01375-f006:**
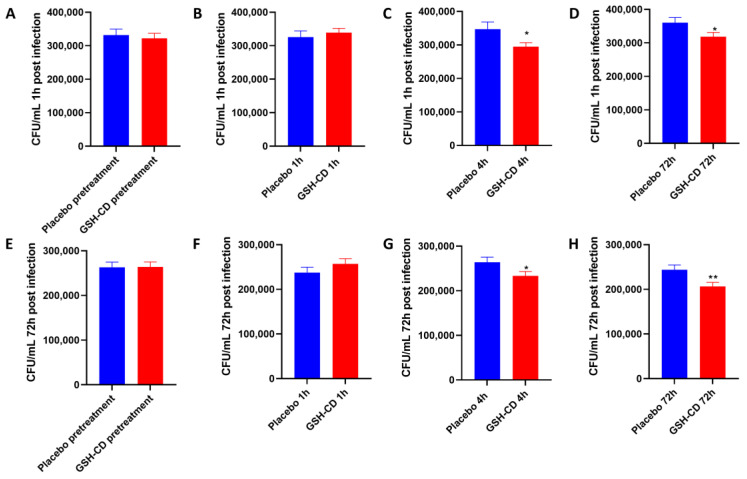
**In vitro *M. avium* survival in whole blood of healthy patients receiving placebo (n = 15) or GSH-CD (n = 15) topical treatment**. (**A**) *M. avium* survival 1 h post-infection in blood drawn from placebo and GSH-CD groups 30 min prior to topical treatment. (**B**) *M. avium* survival after 1 h post-infection in blood drawn from placebo and GSH-CD 1 h post-initial topical application. (**C**) *M. avium* survival 1 h post-infection in blood drawn from the placebo and GSH-CD 4 h post-initial topical application (**D**) *M. avium* survival 1 h post-infection in blood drawn from placebo and GSH-CD at 72 h post-initial application. (**E**) *M. avium* survival after 72 h incubation in blood drawn from the placebo and GSH-CD 30 min prior to initial application. (**F**) *M. avium* survival after 72 h incubation in blood drawn from the placebo and GSH-CD at 1 h post-initial application. (**G**) *M. avium* survival after 72 h incubation in blood drawn from placebo and GSH-CD 4 h post-initial application. (**H**) *M. avium* survival after 72 h incubation in blood drawn from placebo and GSH-CD at 72 h post-initial application. All comparisons were made via an unpaired *t*-test with Welch’s correction. A single asterisk (*) indicates statistical significance with a *p*-value < 0.05. Double asterisks (**) indicate statistical significance with a *p*-value < 0.005.

**Figure 7 antioxidants-12-01375-f007:**
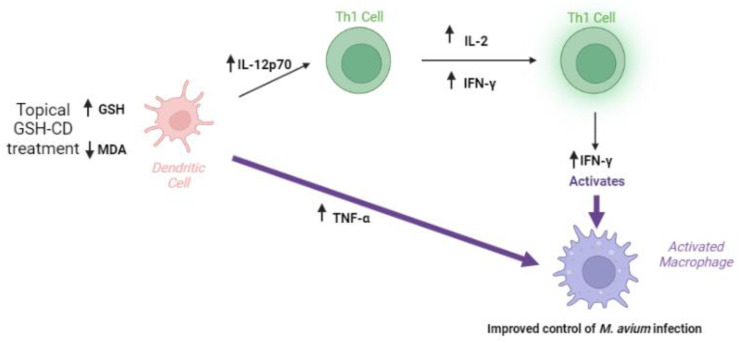
**Summary of clinical trial findings.** The topical application of GSH-CD is associated with increased rGSH in the RBCs and PBMCs of healthy patients and decreased levels of the oxidative stress marker MDA. Increased GSH levels are associated with increased levels of Th1 plasma cytokines IL-12, IL-2, IFN-γ, and TNF-α promoting the activation of macrophages to enhance *M. avium* clearance in whole blood.

## Data Availability

All data generated or analyzed during this study are included in this published article.

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
