# Peer review of "Topical Absorption of Glutathione–Cyclodextrin Nanoparticle Complex in Healthy Human Subjects Improves Immune Response against *Mycobacterium avium* Infection"

_antioxidants, 2023, doi:10.3390/antiox12071375_

Round 1
Reviewer 1 Report
I agree with the final conclusion: Topical application of glutathione cyclodextrin complex is a non-invasive route of administration for GSH delivery with no reported side effects. Despite oral administration is also effective, as the authors report, this is a further step with some advantage for people with digestive intestinal disorders. However, there are some concerns about this article.
1) Data about GSH doses (or alternatively GSH charge in the cyclodextrin encapsulation) and cyclodextrin type are totally missing, but they are essential. The application of 4 sprays of GSH-CD on the ventral abdomen of participants is a rather qualitative description. This is unacceptable in a scientific paper submitted to antioxidants.
Chemical composition of the cyclodextrin charge in glutathione (% w/w, reduced/oxidized GSH ratio and GSH/ Ascorbate ratio inside the capsule and type of cyclodextrin should be given.
2) About data management and statistical significance. According to the protocols (section 2.2 Subject Selection) and Figure 1, 15 patients and 15 placebo participants with appropriate exclusion criteria were selected. However, too many and variable other exclusions are introduced later. For instance, at line 178 (reduced and total GSH measurements) 3 samples of each group were excluded. In other lines (181-182, 190, 194, 200), referred to other determinations (MDA, cytokines), 3, 6, 4, 1 and 2 placebo or patient samples were excluded. As the general exclusion criteria were previously applied, the reasons of so many and variable exclusions would be given for getting a higher confidence in the statistical analysis. Are those samples always obtained from the same individuals? Is there any chance that the excluded samples were uncorrelated and out of range values? Please, clarify these points introducing as much details as possible concerning this point.
3) Figures 3 to 6: Why do authors use the term glutaryl?? First, glutaryl is a radical derived from glutaric acid. Glutaryl and Glutathione are different molecules. Secondly, glutaryl is the trademark (at least from Auro wellness) of a commercial spray that delivers glutathione. Note that it this declared that that Auro Pharmaceuticals Inc. manufactured and suppled GSH-CD consisting of GSH cyclodextrin complex. The manuscript refers to this firm at the Acknowledgments and Conflicts of Interest, but in my opinion, this is not enough, and the use of term Glutaryl should not be used.
Other points to be eventually addressed
PBMCs and RBCs are well-known abbreviations, but anyway abbreviation at the abstract are not recommended. These terms would be defined at that section.
Concerning Figure 3. Why the two panels are referred to reduced and total glutathione, without /with protein normalization? For comparison, it would be convenient the homogenization of the data using the same parameters.
Line 331: According to the increase in TNFa, apoptosis is higher in sprayed patients than in placebo persons even before the bacterial infection. What is the rationality of that response? It would be helpful to introduce a short comment at the discussion to account for this apparently controversial finding.
According to the instruction guidelines, Figures should be distributed throughout the manuscript and inserted close to the related paragraph.
Author Response
Point-by-Point Response to Reviewers Comments
Reviewer 1
Dear Reviewer#1:
Thanks for the constructive feedback and for all your suggestions to enhance the strength and quality of our manuscript. We incorporated all your recommendations in the revised manuscript. We look forward to publishing these important clinical trial findings. Appreciate your time and support.
Comment 1: Data about GSH doses (or alternatively GSH charge in the cyclodextrin encapsulation) and cyclodextrin type are totally missing, but they are essential. The application of 4 sprays of GSH-CD on the ventral abdomen of participants is a rather qualitative description. This is unacceptable in a scientific paper submitted to antioxidants.
Chemical composition of the cyclodextrin charge in glutathione (% w/w, reduced/oxidized GSH ratio and GSH/ Ascorbate ratio) inside the capsule and type of cyclodextrin should be given.
Response 1: Thank you for your suggestion. The concentration of the rGSH in GSH-CD used in the study is 200mg/mL. The applied amount is 0.5 ml (or 100mg rGSH) in 4 sprays. We revised the manuscript to include this information. However, we are unable to report the specific composition of the cyclodextrins as the formulation is a proprietary trademarked product.
Comment 2: About data management and statistical significance. According to the protocols (section 2.2 Subject Selection) and Figure 1, 15 patients and 15 placebo participants with appropriate exclusion criteria were selected. However, too many and variable other exclusions are introduced later. For instance, at line 178 (reduced and total GSH measurements) 3 samples of each group were excluded. In other lines (181-182, 190, 194, 200), referred to other determinations (MDA, cytokines), 3, 6, 4, 1 and 2 placebo or patient samples were excluded. As the general exclusion criteria were previously applied, the reasons for so many and variable exclusions would be given for getting a higher confidence in the statistical analysis. Are those samples always obtained from the same individuals? Is there any chance that the excluded samples were uncorrelated and out of range values? Please, clarify these points introducing as much details as possible concerning this point.
Response 2: We thank the reviewer for bringing up this issue. The excluded samples in each assay possessed out of range values and were not included in the analysis. The excluded values were not obtained from the same individuals and were a result of experimental error. We edited Figure 1 to more accurately demonstrate this data exclusion before the analysis was conducted.
Comment 3: Figures 3 to 6: Why do authors use the term glutaryl?? First, glutaryl is a radical derived from glutaric acid. Glutaryl and Glutathione are different molecules. Secondly, glutaryl is the trademark (at least from Auro wellness) of a commercial spray that delivers glutathione. Note that it is declared that Auro Pharmaceuticals Inc. manufactured and supplied GSH-CD consisting of GSH cyclodextrin complex. The manuscript refers to this firm at the Acknowledgments and Conflicts of Interest, but in my opinion, this is not enough, and the use of term Glutaryl should not be used.
Response 3: Thank you for this suggestion. Glutaryl is the brand name of the GSH-CD complex used in this study. We agree that it is not appropriate to refer to the brand name in the manuscript and was included as a labeling error, as the rest of the manuscript refers to the compound as GSH-CD and not Glutaryl. We revised the figures to include GSH-CD instead of Glutaryl throughout the manuscript.
Comment 4: PBMCs and RBCs are well-known abbreviations, but anyway abbreviations at the abstract are not recommended. These terms would be defined at that section.
Response 4: We thank the reviewer for bringing this issue to our attention. We have fixed the abbreviations in the abstract and added the proper terminology where necessary.
Comment 5: Concerning Figure 3. Why are the two panels referred to reduced and total glutathione, without /with protein normalization? For comparison, it would be convenient the homogenization of the data using the same parameters.
Response 5: Thank you for your question. The GSH levels in the RBCs and PBMCs have been normalized against total protein and are measured as umol/ug protein. The RBC graph contains labeling errors as RBCs were measured as umol/ug protein. The RBC rGSH levels from our collected samples without protein normalization range approximately from 600 uM - 1600 uM. The values on the PBMC graph were also erroneously mislabeled as total GSH and are reduced GSH levels. We repeated the GSH assays on the PBMCs to include n=13 in the GSH-CD arm and included a new analysis. The graphs have been revised to fix these errors.
Comment 6: Line 331: According to the increase in TNFa, apoptosis is higher in sprayed patients than in placebo persons even before the bacterial infection. What is the rationality of that response? It would be helpful to introduce a short comment at the discussion to account for this apparently controversial finding.
Response 6: Thank you for the question and suggestion. TNFa is a pleiotropic cytokine that contributes to host protective effects in the context of mycobacterial infections. Mycobacterium encodes virulence factors such as ESX-1 that prevent phagocytic degradation by macrophages [PMID: 18503637, PMID: 27375559]. TNF-a promotes macrophage phagolysosome fusion and maturation needed to reduce M.tb growth and dissemination of M.tb [PMID: 34755600].GSH has been observed to stimulate the production of Th1 cytokines in the absence of infection in a macrophage cell line including TNF-a [PMID: 31540482],
Conversely, the subject of debate lies in that TNFa also serves as a biomarker for disease and brain dysfunction such as Alzheimer’s and Parkinson’s disease [PMID: 27697064]. Furthermore, TNF-α inhibitors have been FDA approved to treat disorders such as, but not limited to, ankylosing spondylitis, hidradenitis, arthritis, psoriasis, and ulcerative colitis [PMID: 29494032.]. These disorders are often a result of a dysregulated cytokine response promoting excessive inflammation and reactive oxygen species (ROS) production causing a detrimental host effect. GSH provides host protection against ROS in the brain, though further studies are needed to assess if long term GSH-induced TNF-a stimulation would have any effect on healthy tissues [PMID: 25661512.]. We included the rationale and included a suggestion to monitor long term GSH induced TNFa stimulation in a future study in the discussion.
Comment 7: According to the instruction guidelines, Figures should be distributed throughout the manuscript and inserted close to the related paragraph.
Response 7: We thank the reviewer for this comment. We have moved the Figures to their appropriate locations and inserted close to their related paragraphs.

Reviewer 2 Report
Dear Editor!
The presented work "Topical absorption of glutathione-cyclodextrin complex in healthy human subjects improves immune response against mycobacterium infection" by Kayvan Sasaninia et al. is an interesting study to elucidate mechanisms of oxidative stress and inflammatory biomarkers in bacterial infection caused by pathogen M. avium infection in vitro.
The solution of the experimental research topic is interesting and original, and undoubtedly, has practical use in clinical experiments. The manuscript is clearly written and technically sound.
In reviewing the work, the authors should pay attention to the following points of the manuscript and clarify the important aspects of this study.
First of all, the authors did not present a significant difference in the points of the previous study regarding the liposomal glutathione fraction and the glutathione-cyclodextrin complex.
What is the leading component of this analysis, as the main properties of cyclodextrins and the possibility of their use in formulations of various dosage forms as additive agents are well represented in the literature?
It is known that cyclodextrins make it possible to increase the solubility of drug substances, their bioavailability and stability of the drug substance. However, anti-inflammatory, sedative, antioxidant and other properties for cyclodextrins are also presented in the literature. Did the authors take this important point into account for their research?
For this experiment, the role of glutathione, the antioxidant properties of which have also been previously shown significantly in different models by other authors and are no longer questionable? The authors do not consider in their work this important aspect of the active ingredient complex of the study drug as cyclodextrin.
Further, the authors should revise part of the discussion paper by presenting elements of innovation of their study, strengths and limitations of this experiment.
The authors should remove repetitive elements of the results of the paper from the discussion and focus on the unexplained points of the study. The role of the antioxidant glutathione is indisputable, but how is this presented in the literature for M. avium infection?
Is there a difference in the antioxidant properties of whole blood and plasma in these studies?
Have the authors done such comparative analyses to clarify the situation of the effect of the pathogen and the glutathione-cyclodextrin complex?
From the number of participants in the experiment, it appears that in some cases up to half of the results (n=6) were excluded from the statistical analysis. Why were these particular data not presented in the scheme and how did this affect the significance of the results? This should have been discussed in more detail.
The work needs to clarify these important points of the study.
Author Response
Point-by-Point Response to Reviewers Comments
Reviewer 2
Dear Reviewer#2:
Thanks for the constructive feedback and for all your suggestions to enhance the strength and quality of our manuscript. We incorporated all your recommendations in the revised manuscript. We look forward to publishing these important clinical trial findings. Appreciate your time and support.
Comment 1: The authors did not present a significant difference in the points of the previous study regarding the liposomal glutathione fraction and the glutathione-cyclodextrin complex.
Response 1: Thank you for the question. While the previous clinical trials assessed the efficacy of L-GSH and utilized the same markers (GSH levels, MDA levels, cytokine profiles, bacterial survival) as this current study, there are significant differences. The past and present studies have differences in the patient populations, the GSH formulations/dosage used, the route of entry, the duration of treatment and bacterium studies. The previous L-GSH studies [PMID: 26133750, PMID: 34150674] evaluated the efficacy of GSH in augmenting the host responses in immunocompromised patients (HIV and T2DM) with impaired GSH synthesizing capabilities whereas our current study is assessing the safety and efficacy of GSH-CD in healthy patients. Treatment duration and route of entry in the previous studies were 3 months of daily oral administration of liposomal GSH (L-GSH) (1260 mg/mL rGSH per day) compared to the 3 days of twice daily topical application of .5 ml (100 mg/mL rGSH) GSH cyclodextrin nanoparticle complex (GSH-CD). Finally, previous clinical trials challenged PBMCs from HIV or T2DM with M.tb or BCG mycobacterial strains, whereas the current study is the first to assess if GSH is efficacious against M. avium strains. Due to different study parameters, it would be difficult to conclusively assess which route of entry or GSH formulation is more efficacious in increasing blood GSH levels and augmenting the host immune response against mycobacterium. Further experiments are needed to compare the different GSH formulations using the same study parameters.
Comment 2: What is the leading component of this analysis, as the main properties of cyclodextrins and the possibility of their use in formulations of various dosage forms as additive agents are well represented in the literature?
It is known that cyclodextrins make it possible to increase the solubility of drug substances, their bioavailability and stability of the drug substance. However, anti-inflammatory, sedative, antioxidant and other properties for cyclodextrins are also presented in the literature. Did the authors take this important point into account for their research?
Response 2: We thank the reviewer for this question. The efficacy of the active ingredient, rGSH, is the leading component of the analysis. We are assessing if cyclodextrin can serve as a carrier for rGSH delivery. Indeed, there is literature that cites antioxidant properties of some cyclodextrin formulations. The placebo in the study was composed of the same cyclodextrin formulation as the GSH-CD complex without rGSH. Any intrinsic antioxidant capacity the cyclodextrin may confer was not detected in our study as placebo treated patients PBMC MDA levels, our marker for oxidative stress, were observed rise by the study endpoint. Furthermore, we observe no significant difference in pretreated placebo treated plasma IFNg. A significant reduction in IL-2 production was observed along with a nonsignficant reduction in IL-12 and TNF-a by the study endpoint in the placebo group compared to pretreatment. Conversely, we observed GSH-CD topical treatment elevated Th1 cytokines and decreased PBMC MDA levels 72 hours post treatment compared to placebo group, suggesting the inclusion of rGSH in the GSH-CD complex is necessary to observe decreased oxidative stress and immunostimulatory effects of the topical compound. These results are included in the Appendix:
Figure A1: Malondialdehyde (MDA) levels healthy patients prior and post topical placebo application. MDA levels in the PBMCs isolated from patients receiving placebo prior (n=15), 4 hours (n=12) and 72 hours (n=12) post initial topical placebo application. Out of range values were excluded from analysis. Measurements for the PBMCs are normalized against total PBMC protein levels. Comparisons were made via ordinary one-way ANOVA. Asterisks indicate statistical significance with p-value < 0.05
Figure A2: Cytokine profile of heathy patients receiving placebo prior and 72 hours post initial topical application (A) Levels of IL-2 in the plasma prior (n=14) and 72 hours (n=11) group post initial application. (B) Levels of IFN-γ in the placebo group plasma isolated prior (n=13) and 72 hours (n=14) post initial application. (C) Levels of TNF-α in the plasma isolated prior (n=15) and 72 hours (n=13) post initial application. (D) Levels of IL-12 in the plasma isolated from pretreatment (n=15) and 72 hours (n=13) post initial application. Out of range values were excluded from out All comparisons were made via unpaired t-test with Welch’s correction. Asterisks indicate statistical significance with p-value < 0.05.
Comment 3: For this experiment, the role of glutathione, the antioxidant properties of which have also been previously shown significantly in different models by other authors and are no longer questionable? The authors do not consider in their work this important aspect of the active ingredient complex of the study drug as cyclodextrin.
Response 3: We thank the reviewer for this question. The role of GSH was analyzed in conjunction with MDA and cytokine data to establish its ability to reduce oxidative stress and stimulate Th1 cytokine production as previously described in the literature. We have also included data from the placebo group in the manuscript to demonstrate the cyclodextrins in the studied formulation alone does not contribute to antioxidant effects (refer to response 2). The primary outcome of this study was to assess GSH-CD ability to clear M.avium infection in vitro.
Comment 4: The authors should revise part of the discussion paper by presenting elements of innovation of their study, strengths and limitations of this experiment.
Response 4: We thank you for your suggestion. The innovation of this study is to assess if topical application of GSH would elicit an immune response against M. avium. Previous studies have demonstrated topical GSH is efficacious in wound healing and skin brightening, but there are no such studies assessing the immune response after topical GSH application. Studies that have attempted to assess the efficacy of rGSH in augmenting the host immune response in controlling M.tb infections utilized liposomal formulations that were orally administered. We are assessing if topical GSH application would confer a similar therapeutic as orally administered L-GSH formulations, expanding the repertoire of delivery methods for rGSH. The limitations of the study are outlined in the last paragraph of the discussion where we acknowledge the small sample size, the limitation of in vitro vs in vivo efficacy, and the limitation of applicability to M.tb infection. We have adjusted the manuscript to highlight the innovation (topical GSH application and GSH efficacy against M. avium) and expanded the limitations as follows:
“Overall, our findings suggest topical GSH-CD application may improve control of M. avium infection by increasing blood rGSH levels, reducing oxidative stress, and inducting a Th1 promoting cytokine response. Sampling errors resulting in the exclusion of participant data from analysis and reducing comparison group size presents a limitation to this pilot study. Thus, Further studies with increased sample sizes are needed to assess optimal the efficacy of dose range of topical GSH-CD in elevating GSH levels in healthy or immunocompromised patients, such as those with HIV or T2DM. Topical GSH-CD treatment did not result in the complete clearance of M. avium infection by the study endpoint, warranting further studies evaluating optimal dose ranges, treatment duration and combination treatment with first line antibiotics. No adverse effects were reported during this study, however, an evaluation for the potential for skin discoloration or heightened TNF-a levels after long term application of GSH-CD is recommended. While M. avium infection in this study served as a model for mycobacterium infections in the blood, additional in vitro studies are needed to assess the efficacy of GSH-CD against M. tb infection. Additionally, placebo-controlled randomized trials with patients with active M. tb infections are needed to confirm the efficacy of topical GSH-CD as an adjunctive therapeutic approach against tuberculosis.”
Comment 5: The authors should remove repetitive elements of the results of the paper from the discussion and focus on the unexplained points of the study. The role of the antioxidant glutathione is indisputable, but how is this presented in the literature for M. avium infection?
Response 5: We thank the reviewer for this suggestion. We revised the discussion to include more background information with M. avium and clarified previous studies attempted to elucidate L-GSH efficacy against M.tb and BCG strains but not M.avium. We have also eliminated redundancies in the discussion that were stated in the results section.
Comment 6: Is there a difference in the antioxidant properties of whole blood and plasma in these studies?
Response 6: We thank the reviewer for this question. While we observed a decrease in the MDA levels in the PBMCs, we did not see a significant difference in the Plasma MDA levels. Unfortunately, we do not have whole blood data to present, but we can certainly perform these assays in a future study.
:
Comment 7: Have the authors done such comparative analyses to clarify the situation of the effect of the pathogen and the glutathione-cyclodextrin complex?
Response 7: Thank you for the question. This is the first study to assess the efficacy of GSH on M.avium and no other comparative analysis can be performed at this point. Additional studies will be performed in the future to characterize the effects of GSH-CD on M.avium grown in the absence of host cells and the effect the M avium has on GSH-CD.
Comment 8: From the number of participants in the experiment, it appears that in some cases up to half of the results (n=6) were excluded from the statistical analysis. Why were these particular data not presented in the scheme and how did this affect the significance of the results? This should have been discussed in more detail.
Response 8: Thank you for bringing up this issue. We have addressed this issue in the methods section and in the results section and provided an explanation why some data were further excluded from the analysis. The n=6 PBMC rGSH values in the GSH-CD arm have been excluded due to experimental error. We agree this is a large fraction of samples to be missing from the final analysis. We repeated the GSH assay on the PBMCs of those 6 participants. rGSH values from 4 of the 6 participants generated readable values and were included in the comparison group. The remaining 2 did not generate GSH levels detectable by the assays, possibly due to low concentration of extracted PBMCs. We have further edited Figure 1 to accurately reflect the subject selection and comparison group size. We revised the new rGSH PBMC graph to include the additional 4 participant rGSH values in the GSH-CD arm. We also acknowledge in the final paragraph that the data exclusion presents a limitation in the conclusiveness of the results and warrants further clinical trials to confirm its efficacy.

Reviewer 3 Report
I have reviewed the paper by Sasaninia et al.
The findings are interesting and promising, but authors need to clearly explain why some data has been removed from the analysis, and such great statistical differences in such a small group of subjects is hard to believe.
Since the topical glutathione preparation involves nanoparticles, this should be stated in the title and in the abstract. The title also needs to be specific about M. avium infection, to avoid it being misleading.
Being that the groups are rather small (n=12) authors need to clearly explain why “Total GSH levels of 3 patients in the placebo group and 6 patients from the GSH-CD group were excluded from the analysis.”
And why “MDA levels in the PBMCs from 3 participants in each 190 arm were excluded from analysis.”
And why for IFN-gamma “with 1 patient from each arm excluded from the analysis.”
And why “TNF-α levels from 206 2 participants in the placebo group were excluded from analysis”
And why “IL-12p70 levels from 2 participants from 210 each arm were excluded from the analysis”
Figure 1 shows n=15 for each group, which does not seem accurate. A more useful figure should show the actual number of subjects being compared and why certain subjects were excluded from the analysis.
The description “whole blood cultures” is wrong. They should just call it “whole blood infection assay”. “Briefly, blood cultures were infected in vitro with processed M. avium” is inaccurate. A blood culture implies that blood is placed in specific agar to evaluate growth of an organism. Here blood is used for an in vitro inoculation. So the sentence should just be “whole blood was infected in vitro with M. avium…”
Y axis for cytokines in Fig 5 should be in pg/mL
Author Response
Point-by-Point Response to Reviewers Comments
Reviewer 3
Dear Reviewer#3:
Thanks for the constructive feedback and for all your suggestions to enhance the strength and quality of our manuscript. We incorporated all your recommendations in the revised manuscript. We look forward to publishing these important clinical trial findings. Appreciate your time and support.
Comment 1: The findings are interesting and promising, but authors need to clearly explain why some data has been removed from the analysis, and such great statistical differences in such a small group of subjects is hard to believe.
Response 1: We thank the reviewer for bringing up this issue. The excluded samples in each assay possessed out of range values and were not included in the analysis. The excluded values were not obtained from the same individuals and were a result of experimental error. We edited Figure 1 to demonstrate this data exclusion more accurately before the analysis was conducted.
Comment 2: Since the topical glutathione preparation involves nanoparticles, this should be stated in the title and in the abstract. The title also needs to be specific about M. avium infection, to avoid it being misleading.
Response 2: We thank the reviewer for the suggestion. We revised the manuscript to include reference to nanoparticles and M.avium in the title and abstract.
Comment 3: Being that the groups are rather small (n=12) authors need to clearly explain why “Total GSH levels of 3 patients in the placebo group and 6 patients from the GSH-CD group were excluded from the analysis.”
And why “MDA levels in the PBMCs from 3 participants in each 190 arm were excluded from analysis.”
And why for IFN-gamma “with 1 patient from each arm excluded from the analysis.”
And why “TNF-α levels from 206 2 participants in the placebo group were excluded from analysis”
And why “IL-12p70 levels from 2 participants from 210 each arm were excluded from the analysis”
Response 3: We thank the reviewer for the suggestion. We have addressed data exclusion and quality assessment in the methods section and in the results section and provided an explanation why some data were further excluded from the analysis process.
Comment 4: Figure 1 shows n=15 for each group, which does not seem accurate. A more useful figure should show the actual number of subjects being compared and why certain subjects were excluded from the analysis.
Response 4: Thank you for your suggestion. We have edited Figure 1 and added the step for data exclusion which was carried out throughout data quality assessment.
Comment 5: “Whole blood infection assay”. “Briefly, blood cultures were infected in vitro with processed M. avium” is inaccurate. Blood culture implies that blood is placed in specific agar to evaluate growth of an organism. Here blood is used for an in vitro inoculation. So the sentence should just be “whole blood was infected in vitro with M. avium…”
Response 5: We thank the reviewer for this suggestion. We eliminated the reference to “blood cultures” and referred to the samples as “infected/inoculated whole blood” in the manuscript.
Comment 6: Y-axis for cytokines in Fig 5 should be in pg/mL
Response 6: Thank you for bringing this issue to our attention. We have added the proper units, pg/mL, to the Y-axis in the Figure.

Round 2
Reviewer 1 Report
The reply letter has addressed most of my comments. Authors have improved the manuscript and some of the concerns are solved.
Anyway, before definitive acceptance, the following minor points should be checked/ corrected:
Figure 1 3 and 4: Please, check the scale at the y-axis. These figures seem to be very small, and the micromoles/microgram of protein ratios are not usual as compared to other similar studies. Alternatively, data could be referred to mg of protein.
Figures A1 and A2 seem to be introduced in the amended version. If so, figures should be renumbered and located in the appropriate paragraph. Otherwise, these figures can be considered as supplementary material.
Other minor format problems should be repaired
It is reasonable that the specific composition of the cyclodextrins cannot be given, as the formulation is a proprietary trademarked product. However, the reply letter expresses that the applied amount is 0.5 ml (100mg rGSH) in 4 sprays. This information should be given at the legend of Figure 1.
Author Response
Dear Reviewer,
Thanks for the constructive feedback.
We have incorporated all your recommendations in the revised version of the manuscript.
Point-by-Point Response to Reviewer Comments
Reviewer 1
Comment 1: Figure 1 3 and 4: Please, check the scale at the y-axis. These figures seem to be very small, and the micromoles/microgram of protein ratios are not usual as compared to other similar studies. Alternatively, data could be referred to mg of protein.
Response 1: Thank you for this comment. We have modified the graphs to report rGSH and MDA levels in ug/mg as requested.
Comment 2: Figures A1 and A2 seem to be introduced in the amended version. If so, figures should be renumbered and located in the appropriate paragraph. Otherwise, these figures can be considered as supplementary material.
Response 2: Thank you for this comment. We have checked again to make sure Figures A1 and A2 are properly placed in the manuscript and it follows the template and instructions.
Comment 3: It is reasonable that the specific composition of the cyclodextrins cannot be given, as the formulation is a proprietary trademarked product. However, the reply letter expresses that the applied amount is 0.5 ml (100mg rGSH) in 4 sprays. This information should be given at the legend of Figure 1.
Response 3: Thank you for this suggestion. We have additionally included this information at the legend of Figure 1.

Reviewer 2 Report
I already checked the revision of this manuscript. I am willing to recommend an acceptance for publication since the authors proved an excellent improvement for their paper. I also believe that the paper is now reaching the highest standards of the journal. Well done and congratulations!
Author Response
Thank you very much for the positive feedback and for endorsing our manuscript.